# EWE: An Agentic Framework for Extreme Weather Analysis

## Abstract

Extreme weather events pose escalating risks to global society, underscoring the urgent need to unravel their underlying physical mechanisms. Yet the prevailing expert-driven, labor-intensive diagnostic paradigm has created a critical analytical bottleneck, stalling scientific progress. While AI for Earth Science has achieved notable advances in prediction, the equally essential challenge of automated diagnostic reasoning remains largely unexplored. We present the Extreme Weather Expert (EWE), the first intelligent agent framework dedicated to this task. EWE emulates expert workflows through knowledge-guided planning, closed-loop reasoning, and a domain-tailored meteorological toolkit. It autonomously produces and interprets multimodal visualizations from raw meteorological data, enabling comprehensive diagnostic analyses. To catalyze progress, we introduce the first benchmark for this emerging field, comprising a curated dataset of 103 high-impact events and a novel step-wise evaluation metric. EWE marks a step toward automated scientific discovery and offers the potential to democratize expertise and intellectual resources, particularly for developing nations vulnerable to extreme weather.

## 1 Introduction

The increasing frequency and intensity of extreme weather events, driven by anthropogenic climate change, pose a significant and escalating threat to global society and ecosystems. A deep understanding of the physical mechanisms and developmental processes behind these events is therefore paramount. Such analysis is not only critical for accurate attribution and improving future forecasts but is also fundamental to comprehending the broader dynamics of our changing Earth system. However, the prevailing approach to this diagnostic analysis remains a deeply labor-intensive endeavor, reliant on a small pool of meteorological experts who apply years of accumulated, specialized knowledge to each case. In an era of accelerating climate change, where such events are becoming commonplace, this manual paradigm is proving unsustainable. A vast and growing number of events remain unanalyzed, leaving critical patterns and insights undiscovered and hindering our collective ability to adapt.

This critical gap in diagnostic analysis arises in part from the prevailing trajectory of AI development in Earth system science, which has prioritized the development of powerful predictive models over tools that can automate and scale scientific understanding. While forecasting models like Pangu-Weather Bi et al. (2023), GraphCast Lam et al. (2023), and FengWu Chen et al. (2023a) have achieved remarkable success, they do not address the fundamental challenge of condensing and reasoning with the vast body of expert knowledge required for diagnostic analysis.Consequently, the equally crucial domain of post-event analysis, which involves understanding the complex atmospheric dynamics and physical mechanisms that precipitate an extreme event, has been largely overlooked. This diagnostic process is fundamental to improving future forecasts, refining climate models, and informing effective disaster mitigation strategies. The emergence of Large Language Models (LLMs) and intelligent agents presents a promising new frontier, yet their application in this specialized domain is hampered by the critical limitation that their vast knowledge is unmoored from the physical reality captured in high-dimensional meteorological datasets. Without the ability to access, process, and reason over observational data, their analytical potential remains untapped.

To bridge this chasm between abstract reasoning and data-driven scientific inquiry, we introduce the Extreme Weather Expert (EWE), the first intelligent agent framework designed specifically for the

Figure 1: EWE identifies the driving factors of extreme events through a human-like reasoning process by progressively retrieving data and use physics-based diagnostic toolkit.

diagnostic analysis of extreme weather events. As illustrated in Fig. 1, EWE operationalizes the workflow of a human expert. For any given event, it first leverages its embedded meteorological knowledge to formulate a structured analytical plan. It then utilizes a specialized toolkit to retrieve and process vast quantities of relevant meteorological data for the specific time and location. Crucially, EWE autonomously generates Python-based visualization tools to transform raw numerical data into interpretable formats, such as synoptic charts and diagnostic plots. These visualizations are then interpreted by a multimodal large language model (MLLM), enabling a holistic analysis that captures global patterns and key atmospheric features, mirroring the diagnostic process of human experts and enhancing the credibility of its findings.

At its core, the EWE framework integrates three synergistic components: a) **Knowledge-Enhanced Planning** guides the analysis workflow and constrains LLM hallucinations by decomposing complex tasks into knowledge-anchored sub-tasks, leveraging Chain-of-Thought Wei et al. (2022); Yao et al. (2023) prompting based on expert examples. b) **Self-Evolving Closed-Loop Reasoning** employs a unified Checker module to verify the success of each executed action, thereby ensuring operational correctness. c) **Meteorological Toolkit** provides a specialized library of functions for meteorological data retrieval, processing, and the computation of canonical diagnostic equations, which guarantees that EWE can effectively design appropriate analytical tools for a given task.

As the first work dedicated to automated extreme weather analysis, this paper makes several key contributions. To validate EWE and provide a fair benchmark for future research, we establish the first comprehensive dataset for this task, curating 103 high-impact extreme weather events from the past decade, sourced from the EM-DAT database and WMO reports and covering all major IPCC AR6 categories. Furthermore, we introduce a novel, LLM-based step-wise evaluation metric that assesses the entire analytical workflow, from code generation to the extraction of key meteorological insights. This provides a granular assessment of an agent's true diagnostic capabilities. Collectively, these contributions not only advance automated scientific discovery but also hold the potential to democratize expertise, providing an accessible intellectual resource for developing nations that disproportionately suffer from extreme weather yet often lack dedicated specialist teams.

## 2 RELATED WORK

**AI for Meteorology.** Recent advancements in meteorological foundation models, such as Pangu-Weather Bi et al. (2023), GraphCast Lam et al. (2023), and Fengwu Chen et al. (2023a), have revolutionized global weather forecasting with unprecedented predictive accuracy. However, these systems are primarily prognostic tools optimized for numerical fidelity, lacking intrinsic mechanisms for causal inference or diagnostic interpretation. To address the need for interpretability, another stream of research has leveraged explainable AI (XAI) techniques for climate science. XAI methods Srinivasan et al. (2020); Happé et al. (2024) have been used to uncover physical mechanisms, but often at the cost of oversimplifying multivariate interactions or requiring significant domain expertise for post-hoc analysis. Recent work has demonstrated their potential in generating severe-weather outlooks Lawson et al. (2025), recognizing cartographic features on weather maps Takasuka et al. (2024), and assisting in data processing and event diagnosis tasks Zhang et al. (2025b). Yet, these models are confined to executing discrete, human-prompted tasks, such as generating analysis code or interpreting pre-selected visualizations, lacking the autonomy to orchestrate an end-to-end physical mechanism inquiry. Our work builds upon these advancements by proposing a novel paradigm that recasts the LLM from a passive tool into an autonomous agent for extreme

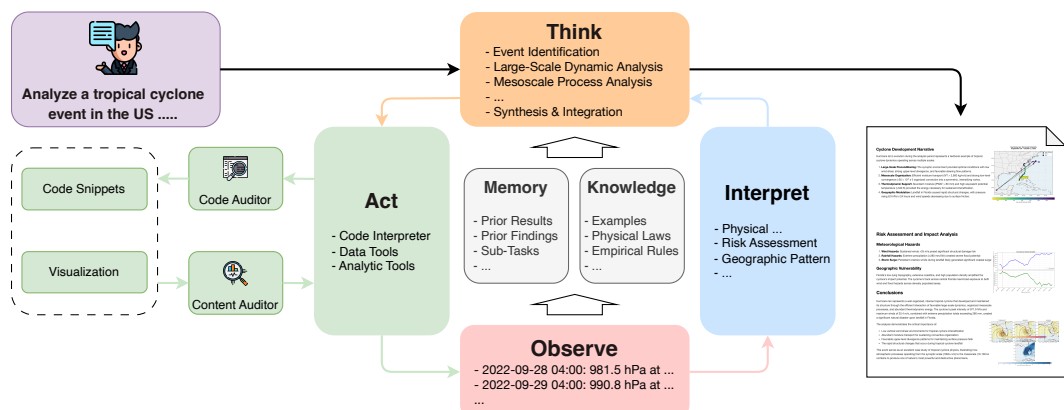

Figure 2: Overview of EWE framework. User request starts self-evolving closed-loop reasoning, and the framework ends with multi-faceted analytical report.

event. We introduce a framework where this agent independently manages the entire workflow, from high-level event definition and exploratory data analysis to the identification of underlying physical mechanisms. This is achieved through a closed-loop, iterative process that directly addresses the limitations of prior work.

**MLLM-Based Agent.** The development of LLM-based agents, which utilize LLMs and increasingly Multimodal Large Language Models as a reasoning backbone, has seen rapid progress Wang et al. (2024a); Xi et al. (2025); Sumers et al. (2023). These agents are typically augmented with memory Zhang et al. (2025c); Zhong et al. (2024), structured planning Wang et al. (2022); Weng et al. (2022), and tool-use capabilities Chen et al. (2024b); Schick et al. (2023); Shen et al. (2023) to interact with external environments and act upon them. Researchers have extensively explored methods to enhance these agents, including sophisticated prompting for reasoning Madaan et al. (2023); Besta et al. (2024), RAG for knowledge extension Gao et al. (2023); Lewis et al. (2020), and reinforcement learning for post-training alignment Bai et al. (2022); Kaufmann et al. (2024). Such enhanced agents have demonstrated significant promise in complex domains like embodied simulation Liang et al. (2022); Song et al. (2023), video games Wu et al. (2024).

## 3 METHOD

### 3.1 OVERVIEW.

Traditional diagnosis of extreme weather events relies on experts manually synthesizing multi-source data to reconstruct an event's physical mechanisms. This manual-centric approach is labor-intensive, time-consuming, poorly scalable, and prone to subjective biases, proving inefficient for rapidly evolving systems like cyclones. To overcome these limitations, we formalize extreme events diagnosis as an autonomous exploration and reasoning task for an MLLM-powered agent. We abstract the diagnostic workflow into an iterative trajectory, $\tau = (t_k, a_k, o_k, i_k)_{k=1}^{N}$, representing a cycle of Thought, Action, Observation, and Interpretation. In this loop, the agent iteratively plans, uses tools on meteorological data, and integrates observations with its internal knowledge to progressively construct a physically consistent causal explanation for the target event. To implement this process, we propose Extreme Weather Expert (EWE), a novel framework integrating three core components as illustrated in Fig. 2:

**Knowledge-Enhanced Planning**. Leverages Chain-of-Thought (CoT) prompting with expert exemplars to decompose the diagnostic task into a plan of knowledge-anchored sub-goals. This guides the agent towards a rigorous and efficient analytical procedure.

**Self-Evolving Closed-Loop Reasoning.** The agent executes the plan by invoking tools. A unified Checker module then validates both the operational success and physical plausibility of each action's output before proceeding, ensuring the integrity of the entire diagnostic pathway.

**Meteorological Toolkit.** A specialized library of functions for meteorological data retrieval, processing, and computation of canonical diagnostic equations. These tools provide the necessary empirical grounding for the agent's scientific conclusions.

## 3.2 KNOWLEDGE-ENHANCED PLANNING WITH CHAIN-OF-THOUGHT

To overcome the limitations of Large Language Models in complex meteorological analysis, such as incomplete reasoning and a high propensity for hallucination, we propose a knowledge-enhanced planning approach. This method utilizes expert-annotated chain-of-thought Wei et al. (2022) to structure the LLM's problem-solving process. We argue that for domain-specific problem-solving, emulating the reasoning path of an expert is a superior strategy for harnessing the internal knowledge of the model and ensuring its reliable application. Therefore, we propose an analytical framework to constrain the reasoning process, guiding the model to effectively retrieve, structure, and apply its latent meteorological knowledge. Specifically, we first manually annotate step-by-step analytical guidelines for different types of extreme events. These guidelines are provided to the agent for initial planning and are also stored in a memory module as persistent and robust context, which the LLM can reference when executing subtasks. This dual mechanism ensures that domain expertise constrains the entire process, from high-level planning to low-level execution.

These CoT guidelines are constructed based on two key principles: **1. Step-by-step Analysis.** This principle decomposes complex diagnostics into a multi-scale, sequential workflow that emulates expert procedures (e.g., identifying circulation patterns, calculating diagnostic fields like potential vorticity). **2. Knowledge Grounding in Reasoning.** This principle mandates that each reasoning step explicitly cites the underlying physical laws, equations, or empirical rules. By forcing the LLM to align its reasoning with this expert-defined structure, it learns to correctly access and utilize its vast but often unstructured internal knowledge.

## 3.3 SELF-EVOLVING CLOSED-LOOP REASONING

The core of our agent is that it operates within a self-evolving, closed-loop reasoning framework to automate the diagnosis of physical mechanisms. This framework enables the agent to iteratively refine its analytical approach by seamlessly integrating thought, action, observation, interpretation, and multi-faceted feedback.

The process begins with a thought phase. At each step $k$, the agent queries its structured memory to reason about the current analytical objective. This process generates a structured reasoning trace, which explicates the logic for formulating the subsequent action, $a_k$. This action, typically a piece of code for data processing or visualization, is then executed in the environment. The agent performs an initial self-debug Chen et al. (2023b); Wang et al. (2024b) based on direct environmental feedback, such as execution errors or returned data, forming the primary feedback loop.

However, for complex physical mechanisms diagnosis, relying solely on environmental feedback is insufficient, as it fails to detect latent flaws that do not cause explicit errors but can lead to erroneous scientific conclusions. To address this, we introduce a dual-auditor module for comprehensive self-evaluation. The Code Auditor ensures the procedural correctness of the agent's generated code. It performs static analysis to identify subtle yet critical bugs that escape standard exception handling, such as the incorrect use of tool parameters or flawed data indexing. By flagging these latent bugs, the auditor prevents the agent from deriving physical insights from corrupted data analysis. The Content Auditor ensures the semantic integrity and perceptual clarity of the generated outputs, particularly visualizations. In atmospheric science, visual analysis is paramount. This auditor assesses visualizations for issues like over-plotting, low-contrast color schemes, or occluded labels that could obscure or misrepresent critical weather patterns.

The feedback from both the environment and the dual-auditor system is then integrated, informing the agent's next thought phase. This closed-loop process allows the agent to continuously evolve its strategy, correcting not only its code but also its data presentation methods. If the current sub-task's goal is met, the agent advances to the next step in its overarching plan. This iterative refinement cycle repeats until a logically coherent diagnosis of the physical mechanism is achieved, at which point the process terminates.

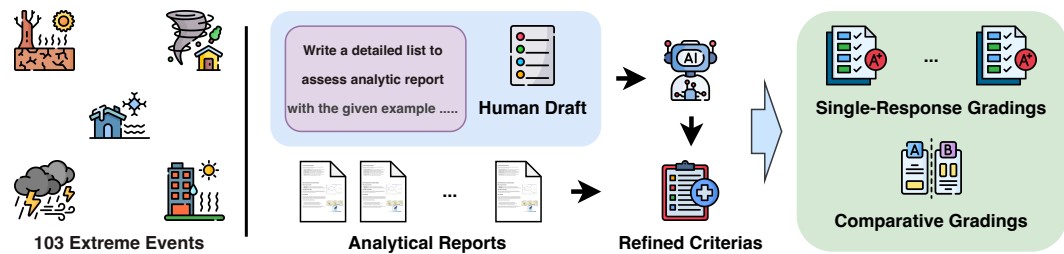

**103 Extreme Events**     **Analytical Reports**     **Refined Criterias**

Figure 3: For all extreme event types, human-drafted checklists are refined by LLM. Then analytical reports generated by EWE are assessed by corresponding refined criteria.

**Memory.** To support this long-horizon reasoning process, the agent employs a dynamic memory management module. This module archives successfully executed actions and their outcomes as positive exemplars to guide future reasoning. Upon the completion of a sub-task, the agent distills the key findings into a concise summary for long-term retention. Concurrently, it prunes all transient information associated with the completed sub-task, such as intermediate variables and detailed logs. This strategy of knowledge distillation and context pruning ensures that the input context remains salient and computationally tractable, preventing performance degradation from excessive context length while reducing token consumption.

### 3.4 Meteorological Toolkit

**Data Acquisition Tool.** We developed data acquisition tools to retrieve multivariate meteorological data for specific extreme weather events. The tool queries a database by a specified temporal range and a list of required variables. For this study, it sources data from the $0.25°$ ERA5 reanalysis dataset ($>300$TB). To provide a crucial baseline for anomaly quantification, the tool also extracts a 30-year climatology. All retrieved data are packaged into NetCDF files with complete metadata preserved, ensuring direct interpretability by our AI agent. The tool features a data-source agnostic design, making it readily adaptable to other reanalysis or observational datasets.

**Analysis Tools.** We introduce the Analysis Toolkit to address a key limitation of LLMs in scientific domains: while proficient at generating code for basic tasks (e.g., plotting with matplotlib), they struggle with complex, domain-specific computations. For instance, calculating Integrated Vapor Transport (IVT)—a critical diagnostic—requires both deep meteorological knowledge and precise coding, a dual challenge for generalist models. Our toolkit directly addresses this by providing a curated library of pre-verified Python functions for such diagnostics. Each function was implemented and rigorously cross-validated by domain experts to ensure scientific accuracy.

## 4 Benchmark

To rigorously assess the agent's proficiency in analyzing extreme events, we have curated a dataset of extreme events and introduce a multi-dimensional, step-wise automated evaluation framework. For a given extreme weather event and an initial user prompt, the interaction between the agent and the environment is captured as a trajectory, $\tau = (\tau_0, \tau_1, ..., \tau_n)$. Each step $\tau_k = (a_k, o_k, i_k)$ comprises an action $a_t$ executed by the agent, the resultant observation $o_t$ from the environment (e.g., numerical results, meteorological visualizations), and a corresponding interpretation $i_t$ generated by the agent for diagnostic-related observations. Our framework evaluates the performance of the agent along three primary dimensions, code, visualization, and physical diagnostic analysis. While the final report is a key artifact,we argue that a step-wise evaluation is crucial for effective credit assignment and for optimizing the decision-making policy.. Therefore, our framework computes a scalar reward $r_k$ for each step, formulated as: $r_k = \mathcal{E}(e, a_k, o_k, i_k)$ where $e$ represents the event context and $\mathcal{E}$ is our automated evaluator. The overall automatic evaluation pipeline is illustrated in Fig. 3.

**Event Collection.** We constructed a dataset of high-impact extreme weather events by curating records from the EM-DAT database and the WMO State of the Global Climate reports. Our methodology prioritizes events from the last decade with significant, documented socio-economic conse-

quences, global coverage (excluding Antarctica), and alignment with IPCC AR6 event categories: temperature extremes (heatwaves, cold waves), extreme precipitation, droughts, and storms (tropical/extratropical cyclones). Events without demonstrable human impact, regardless of meteorological severity, were excluded. Each event sample is annotated with its precise start and end dates, location, and type. Missing temporal information was manually verified using public sources. To ensure accurate alignment with local time, the timezone for each event location was recorded, facilitating analysis based on the natural day.

**Checklist Annotation and Generation.** We introduce a systematic evaluation framework centered on LLM-generated checklists to standardize the assessment of extreme event analysis. Initial explorations with zero-shot generation yielded checklists that lacked the required granularity and contextual relevance for specific events and analytical tasks. To overcome this, we developed a one-shot exemplar-guided prompting strategy. The core of our method is a meticulously hand-crafted exemplar, which serves as a one-shot prompt. This exemplar details a multi-modal evaluation rubric for a canonical task, defining precise scoring criteria for code, visualizations, and physical diagnostics. By conditioning on this high-quality exemplar, the LLM learns to generate checklists that are both structurally sound and semantically aligned with the task. The fidelity of these generated checklists is further ensured through a rigorous human verification process on a randomly sampled subset, validating their accuracy, coherence, and comprehensiveness.

The evaluation rubric itself is multi-dimensional. It first assesses Code Fidelity, verifying the correctness of the implementation, particularly the data processing pipelines, visualization rendering code, and the accurate formulation of fundamental physical equations, such as potential vorticity. The rubric then evaluates Visualization Quality, assessing the informativeness and perceptual clarity of the visual outputs, including the efficacy of color maps, the precision of legends, and the clarity of all textual annotations. Most critically, it gauges the Depth of Physical Interpretation. Beyond factual accuracy and consistency with visualizations, we evaluate the analysis's explanatory power. We prioritize a coherent, physically-grounded narrative that establishes clear causal links. For instance, instead of merely stating an observation such as "positive vorticity advection", a high-quality analysis must causally link this observation to its physical implications, such as inducing upward vertical motion and contributing to cyclogenesis. The objective is to reward a demonstrated understanding of the underlying dynamics, rather than the simple recitation of disconnected diagnostic facts.

**Step Classification.** Each interaction step within the agent's trajectory is assigned to a specific behavioral category (e.g., Data Exploration, Event Characterization, Dynamics Analysis). Technically, this is achieved by first constructing a textual representation for each step, which concatenates the action, observation, and the interpretation. This textual block is then processed by a Large Language Model, which leverages its understanding of the semantic context to classify the step according to a pre-defined taxonomy. This methodology facilitates the automated, granular annotation of agent-environment interaction traces, which is a prerequisite for our subsequent category-aware evaluation, where distinct sets of criteria are applied to assess the quality of different types of steps.

**Evaluation** Our methodology for evaluating the quality of generated content relies on a robust, MLLM-based framework. For each classified step of the analytical process, we aggregate all pertinent artifacts—code, visualizations, and textual interpretations—along with the source event data and a structured checklist. This consolidated multimodal content is then submitted to an MLLM-based evaluator, which is primed with specialized meteorological domain knowledge for assessment.

To mitigate potential evaluator bias and ensure a fair comparison, we employ two state-of-the-art models, namely GPT-4.1 and Gemini-2.5-pro as judge. The evaluation is conducted under two complementary protocols:

**Single-Response Grading:** Each step of agent is assessed in isolation to assign an absolute score reflecting its intrinsic quality. This protocol measures the standalone performance of each agent.

**Comparative Grading:** For a given event, outputs from all competing models are presented concurrently to the judge. The judge first identifies the most meritorious response to serve as an ad-hoc standard, and subsequently grades all candidates based on their quality relative to this standard.

## 5 EXPERIMENTS

### 5.1 EXPERIMENTAL SETTINGS

Table 1: Quantitative Evaluation of LLMs on the Benchmark. Scores are assigned by an mLLM-as-a-judge for seven key stages: analysis planning, data exploration, event identification, synoptic-scale analysis, mesoscale analysis, thermodynamic analysis and final report. The evaluation is conducted in two modes: Single-Response Grading (SG) and Comparative Grading (CG). Scores are normalized to [0, 1], with higher values indicating better performance.

| Model | Plan | Data | Identification | Synoptic. | Mesoscale. | Thermo. | Report |
|---|---|---|---|---|---|---|---|
| *Group: SG* | | | | | | | |
| Gemini-2.5-Pro | 0.898 | 0.370 | 0.650 | 0.779 | 0.485 | 0.657 | 0.839 |
| Claude-4-Sonnet | 0.838 | **0.800** | **0.783** | 0.758 | **0.700** | **0.667** | **0.981** |
| GPT-4.1-2025-04-14 | **0.947** | 0.783 | 0.720 | **0.785** | 0.670 | 0.658 | 0.828 |
| Llama-4-Maverick | 0.669 | 0.729 | 0.452 | 0.530 | 0.343 | 0.396 | 0.587 |
| o4-mini-2025-04-16 | 0.974 | 0.751 | 0.607 | 0.780 | 0.655 | 0.661 | 0.720 |
| *Group: CG* | | | | | | | |
| Gemini-2.5-Pro | 0.789 | 0.441 | 0.658 | 0.731 | 0.483 | 0.606 | 0.787 |
| Claude-4-Sonnet | 0.797 | 0.601 | **0.832** | **0.837** | **0.782** | **0.750** | **0.950** |
| GPT-4.1-2025-04-14 | **0.832** | **0.624** | 0.750 | 0.776 | 0.664 | 0.696 | 0.869 |
| Llama-4-Maverick | 0.686 | 0.602 | 0.485 | 0.477 | 0.335 | 0.357 | 0.486 |
| o4-mini-2025-04-16 | 0.787 | 0.514 | 0.646 | 0.816 | 0.652 | 0.720 | 0.421 |

**MLLMs**. The task of agent demands strong reasoning, long-context processing, and multi-modal understanding. We therefore selected a range of state-of-the-art MLLMs, including open-source (LLAMA-4-Maverick) and proprietary (Claude-4, GPT-4.1, Gemini-2.5-Pro, O4-Mini) models. For all experiments, we set the inference temperature to 0 for reproducibility. Each agent run was limited to 40 steps, beyond which the task was marked as a failure. Code and figure auditor were performed by the same model instance.

**Evaluation.** We utilized GPT-4.1 for both step classification and a checklist-based content evaluation. The evaluator was supplied with

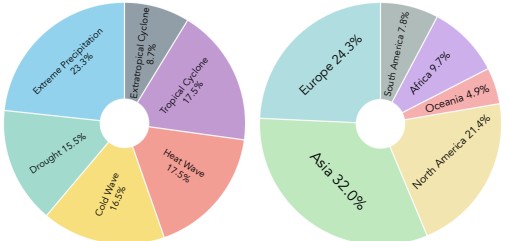

Figure 4: Statistics of extreme events in the test set: the left panel shows the distribution of event categories, and the right for the geographical distribution.

metadata (variables, units, time) for accurate assessment. While we tested Gemini-2.5-Pro as an alternative, it tended to penalize correct code, leading to inaccurately low scores. Consequently, we designated GPT-4.1 as the sole judge for our evaluation to ensure the consistency and reliability of our results.

**Dataset**. Our dataset contains 103 extreme weather events across six types (Fig. 4). The most prominent categories are cyclonic events (26.2% combined) and extreme precipitation (23.3%). The dataset offers global coverage across all inhabited continents, with a distribution skewed towards Asia (32.0%), Europe (24.3%), and North America (21.4%), reflecting global population and data availability patterns.

## 5.2 MAIN RESULTS

We present a comprehensive evaluation of five leading Large Language Models on a complex, multi-step meteorological analysis task. The performance of each model is quantified using an MLLM-as-a-judge Chen et al. (2024a); Zhang et al. (2025a) across seven distinct stages of the analysis pipeline. The results, presented in Tab. 1, detail the performance of five state-of-the-art models across seven distinct analytical stages under two evaluation settings: Single-Response Grading (SG) and Comparative Grading (CG).

**Performance in Single-Response Grading (SG) Evaluation.** Our independent SG evaluation reveals significant task specialization among models (see Tab. 1 ). For instance, while o4-mini-2025-

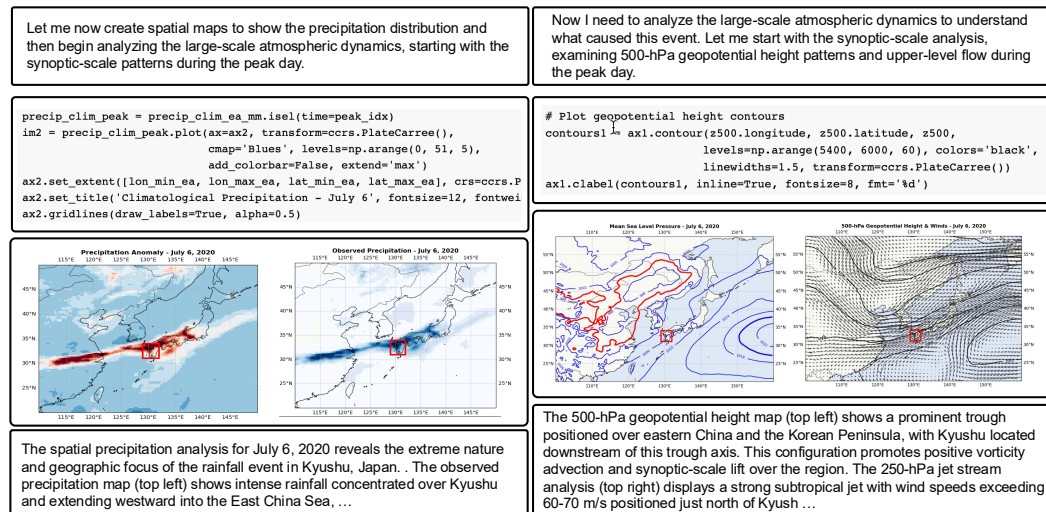

Figure 5: Workflow example of extreme precipitation event analysis. Each step, from top to bottom, presents the agent's thought, action, observation, and interpretation.

04-16 excels at the initial Plan stage (0.974), Claude-4-Sonnet dominates the subsequent analytical phases, achieving the highest scores in event identification (Identi., 0.783), mesoscale analysis (Meso., 0.700), and final synthesis (Report, 0.981). GPT-4.1-2025-04-14 also shows strong performance, particularly in synoptic-scale analysis (Synop., 0.785).

**Performance in Comparative Grading (CG) Evaluation.** Our results show that Comparative Grading (CG) is a more discriminative evaluation protocol than Single Grading (SG). Under CG, the performance gaps between models widen significantly, particularly in complex analytical stages. For instance, the performance spread between the top and bottom-performing models increased from 0.255 (SG) to 0.360 (CG) in the Synoptic stage, and from 0.394 to 0.529 in the Report generation stage. This suggests that while most models produce acceptable standalone outputs, direct side-by-side comparison effectively surfaces nuanced qualitative differences. This sharpened evaluation clarifies the model hierarchy. Claude-4-Sonnet emerges as the top performer, excelling in a majority of analytical stages: Identi. (0.802), Synop. (0.827), Meso. (0.782), Thermo. (0.750), and Report (0.950). Its superior scientific reasoning is further highlighted by its scores increasing in several tasks under CG, against a general deflationary trend. GPT-4.1-2025-04-14 remains the strongest in upstream planning and data processing tasks (Plan: 0.832, Data: 0.624), whereas models like Llama-4-Maverick show a more significant performance drop, confirming CG's efficacy in identifying model-specific strengths and weaknesses.

**Examples of Qualitative Results.** Fig. 5 illustrates the workflow for analyzing extreme precipitation events, which includes two main stages: event qualification and large-scale weather analysis. For better understanding, each stage is organized from top to bottom to show the agent's plan, the code generated to accomplish the plan, the resulting visualization, and the analysis of the results. These steps correspond to the thought, action, observation, and interpretation components described in the main text. This structure clearly demonstrates how the agent decomposes the problem and performs end-to-end reasoning and analysis.

## 5.3 ABLATION STUDY

We conduct comprehensive ablation studies on a limited set of samples. Our framework integrates three key components: analysis tools, a code and figure auditor for verifying and correcting the generated code and analysis logic; and a meteorological Chain-of-Thought (CoT) that infuses domain knowledge into reasoning process. As shown in Tab. 2, we evaluate the model performance on three distinct analysis dimensions: synoptic-scale, mesoscale, and thermodynamic analysis.

**Full model vs. Baseline.** Our full model achieves the best performance across all three metrics. Compared to the baseline model which excludes all components, the full model demonstrates a significant improvement, particularly on synoptic-scale (+0.239) and mesoscale (+0.213) analysis. This underscores the substantial value of the synergistic integration of our modules.

**Importance of Analysis Tools.** Removing the tools module from the full model results in a marked performance degradation across all metrics, most notably in thermodynamic analysis, where the score plummets from 0.679 to 0.537. This is expected, as thermodynamic analysis (e.g., calculating potential temperature) heavily relies on precise numerical computations, which is the core function of the tools module. This highlights that equipping the model with external computational tools is indispensable for tackling scientific analysis tasks.

**Role of the Auditor.** Similarly, removing the auditor module leads to a performance drop, especially in mesoscale analysis. This indicates that the auditor plays a critical role in ensuring the correctness of the figure and the reliability of the generated code. Through self-auditing and correction, the model produces more accurate and reliable analyses, mitigating performance losses caused by faulty code or unclear visualization.

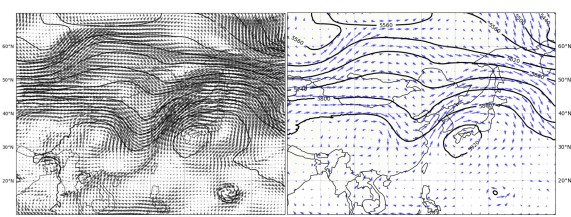

Figure 6: Visually analysing the contributions of content auditor. See text for details.

Fig. 6 highlights the contribution of our Figure Auditor to visualization clarity. The baseline generation (left), depicting the 500 hPa geopotential height and wind, is characterized by a high-density wind vector field that creates visual clutter and obscures synoptic-scale features. In contrast, the refined output guided by our auditor employs a sparsified vector field. This strategic downsampling significantly enhances perceptual clarity, allowing for immediate identification of large-scale circulation. A clear example is the pronounced low-pressure trough, where the refined visualization makes the southward-dipping contours and the associated pre-trough southwesterly and post-trough northwesterly flows immediately apparent. This demonstrates the auditor's effectiveness in producing scientifically interpretable visualizations.

**Foundational Role of CoT.** Although there is no setting where only the cot is removed, the starkly poor performance of the baseline model serves as strong evidence for the foundational role of cot. Without the meteorological domain knowledge and structured reasoning pathways provided by cot, the model struggles to effectively utilize the tools, even when available, leading to a comprehensive collapse in analytical capability.

Table 2: Ablation study on the effectiveness of key components.

| Tools | Auditor | CoT | Synop. | Meso. | Thermo. |
|---|---|---|---|---|---|
| × | ✓ | ✓ | 0.752 | 0.619 | 0.537 |
| ✓ | × | ✓ | 0.768 | 0.636 | 0.665 |
| × | × | × | 0.548 | 0.467 | 0.502 |
| ✓ | ✓ | ✓ | 0.787 | 0.680 | 0.679 |

## 6 CONCLUSION

This study proposes Extreme Weather Expert (EWE), the first intelligent agent for extreme weather event analysis, addressing limitations of traditional labor-intensive methods. EWE integrates three core components—Knowledge-Enhanced Planning, Self-Evolving Closed-Loop Reasoning, and a Meteorological Toolkit—to enable credible, end-to-end diagnosis. A 103-event dataset and LLM-based stepwise evaluation framework are built for validation. Experiments show Claude-4-Sonnet's superiority in key stages, and ablation studies confirm EWE components' necessity. EWE establishes a foundational paradigm for autonomous extreme weather diagnosis, offering potential avenues for extension—such as integrating real-time observational data streams or expanding support for underrepresented extreme weather types. This research not only advances the application of AI agents in meteorological science but also provides a practical tool to mitigate the socio-economic impacts of extreme weather events, particularly in regions with limited data and expert resources.

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

# A APPENDIX

## A.1 USAGE OF LANGUAGE MODELS

We used a large language model (LLM) during the preparation of this manuscript, solely for editorial purposes such as correcting typographical errors, improving grammar, and enhancing clarity and readability.

## A.2 EXAMPLES OF BENCHMARK

```yaml
- name: 2018 North-east Asia heatwave
  region: Eastern Asia
  subregion: Japan
  event: extreme heat
  start: '20180701'
  end: '20180715'
  timezone: Asia/Tokyo
- name: February 2021 Texas Cold Wave
  region: Northern America
  subregion: United States of America
  event: cold wave
  start: '20210213'
  end: '20210217'
  timezone: America/Chicago
- name: "2021 China extreme precipitation"
  region: Eastern Asia
  subregion: China Henan
  event: "extreme rainfall"
  start: '20210717'
  end: '20210724'
  timezone: Asia/Shanghai
```

