# OpenReview forum: "EWE: An Agentic Framework for Extreme Weather Analysis"
_ICLR.cc/2026/Conference — Submitted to ICLR 2026_

### Official Review · Reviewer_ioDy · 2025-10-29

**Soundness:** 1
**Presentation:** 1
**Contribution:** 1
**Rating:** 2
**Confidence:** 3

**Summary:**

This manuscript introduces an intelligent LLM agent to emulate expert knowledge in analyzing weather and climate extreme events. Given an event of interest and related information for it, the LLM agent outputs prompts, code snapshot and visualizations to analyze the extreme event.
The manuscript provides also a benchmark as a collection of extreme events from databases like e EM-DAT and WMO. Experiments were conducted with ERA5 and different LLM agents.

**Strengths:**

- It is promising to use LLM for extreme event analysis.
- The manuscript introduces a benchmark useful for studies of extreme weather and climate events.

**Weaknesses:**

- Unfortunately, I couldn't find the introduced benchmark? Benchmarks should be documented and presented to the reviewers. Otherwise, it cannot be checked.
- Line 245: this is not true. The framework is not data source agnostic since there are no other experiments other than on ERA5.
- Although it is mentioned in L285-287 that human verification is included, it is unclear how the validation is conducted e.g., are there experts from extreme event attribution? As I understood from the paper there is no human judge to provide ground truth but rather ChatGPT is used. I would rather rely on experts in the corresponding field rather than on LLM to provide ground truth for a benchmark. The LLM as I understood is the main source of the ground truth analysis.
- The paper lacks a lot of details related to methodology and the writing is not clear.
- The manuscript does not discuss any limitations.

**Questions:**

- Benchmark in keywords is missing.
- Line266: what is the event context $e$?
- Table 2: which model is being used here?

**Minor**:
- L42: Format citation in parentheses
- L44: Typos: Consequently.
- Line95: these are not foundation models! Foundation models are e.g., to name a few works [WeatherGenerator](https://www.ecmwf.int/en/about/media-centre/news/2024/weathergenerator-project-aims-recast-machine-learning-earth-system), [AtmoRep](https://arxiv.org/abs/2308.13280),  and [Aurora](https://www.nature.com/articles/s41586-025-09005-y).

---

### Official Review · Reviewer_v5ef · 2025-10-31

**Soundness:** 2
**Presentation:** 3
**Contribution:** 2
**Rating:** 4
**Confidence:** 4

**Summary:**

This paper aims to address a critical bottleneck in climate science: the diagnostic analysis of extreme weather events. The authors argue that the current paradigm, which relies on manual analysis by domain experts, is inefficient and unscalable, posing a significant obstacle to scientific progress in an era of accelerating climate change. To meet this challenge, the authors introduce the Extreme Weather Expert (EWE), which they present as the first intelligent agent framework dedicated to automating this diagnostic task. EWE is designed to emulate the workflow of a human expert by integrating knowledge-guided planning, tool use for data retrieval and visualization, and multimodal interpretation of meteorological data.
The EWE framework is built on three core pillars: (a) knowledge-enhanced planning using expert exemplars and Chain-of-Thought (CoT) prompting; (b) self-evolving closed-loop reasoning with a dual "auditor" module for code and content; and (c) a specialized meteorological toolkit for data access and scientific computation.
The authors claim three primary contributions: (1) the EWE framework itself; (2) a new benchmark dataset of 103 high-impact extreme weather events; and (3) a novel, LLM-based step-wise evaluation metric for assessing agent performance on this task.

**Strengths:**

S1: The problem addressed by this paper is of significant scientific and societal importance. The shift in focus within AI for Earth Science from pure prediction (the domain of models like Pangu-Weather and GraphCast) to automated scientific understanding and diagnosis is a crucial and welcome research direction. This work pioneers a new task definition: automated post-hoc diagnostic reasoning. This is a valuable contribution in itself, as it frames a complex scientific workflow as a tractable problem for modern AI agents.

S2: A primary strength of the EWE agent is its end-to-end nature. It attempts to cover the entire scientific discovery process, from formulating a plan and retrieving raw data to generating visualizations and finally interpreting them to form a scientific narrative. This holistic approach is more ambitious than systems that focus on discrete sub-tasks. Furthermore, the inclusion of a specialized meteorological toolkit with pre-verified functions for complex diagnostic calculations (e.g., Integrated Vapor Transport) is a thoughtful and highly practical design choice. It correctly identifies a key weakness of general-purpose LLMs in domain-specific scientific computation.

S3: The creation of a new, curated benchmark dataset, even a modest one (103 events), is a vital contribution. Such resources are essential for catalyzing research in the field, ensuring reproducibility, and providing a basis for fair comparison of future methods. Similarly, the proposal of a step-wise evaluation metric is a valuable idea. Assessing the entire reasoning trajectory, rather than just the final report, allows for more granular credit assignment and represents a more mature approach to evaluating agent performance.

**Weaknesses:**

W1: The paper's entire quantitative evaluation (Table 1) relies on a single multimodal large model, GPT-4.1, as the judge. This "LLM-as-a-Judge" approach is widely documented to suffer from multiple potential biases, such as verbosity bias (favoring longer answers), position bias, and overlooking fallacies in reasoning. The paper's core claims rest on an evaluation protocol that has not been sufficiently validated, constituting a fatal flaw in its scientific rigor. Specifically, the paper suffers from: (1) Lack of human-in-the-loop calibration: The authors provide no validation of their judge model against human expert evaluations (e.g., Spearman's rank correlation), leaving the external validity of the scores in question. (2) Lack of heterogeneous judge consistency measurement: The paper dismisses the Gemini-2.5-Pro model on the grounds that it "tended to penalize correct code", a non-scientific justification that highlights the fragility of the evaluation. A robust benchmark should report inter-judge reliability across multiple judges from different providers.

W2: The core components of the EWE agent—Chain-of-Thought (CoT) for planning, tool use, and a feedback/auditor loop—are standard paradigms in contemporary LLM agent research. The paper fails to sufficiently articulate how its specific implementation or integration of these components constitutes a novel technical contribution to the field of AI agents. The primary novelty of the paper lies in the application to a new scientific domain, not in a new agent architecture. The current framing (e.g., "the first intelligent agent framework" in the abstract) overstates the architectural innovation. Works such as ReAct, Toolformer, and self-debugging have already established these concepts. The paper lacks direct comparisons or ablation studies against these baseline paradigms to isolate and demonstrate the superiority of a novel architectural component. The contribution should therefore be framed as a successful system integration and application rather than a fundamental breakthrough in agent design.

W3: The paper uses the term "diagnostic analysis" to describe its task of uncovering "physical mechanisms". In meteorology, this is closely related to a well-established and methodologically rigorous field known as "extreme event attribution". The paper fails to engage with this body of literature or clarify how its task differs from formal attribution studies. EWE performs automated synoptic analysis, which is often a preliminary step in a formal event attribution study, but the two are not equivalent. Formal attribution typically requires comparing observed facts with counterfactual climate model simulations (e.g., a world without climate change) to quantify the influence of anthropogenic activity. EWE does not perform this kind of counterfactual comparison. While the paper links its motivation to "attribution," it fails to clarify this relationship in its related work, and this terminological imprecision could be viewed by domain experts as a sign of academic immaturity or over-claiming.

**Questions:**

Q1: Can the authors provide a validation of their GPT-4.1 judge? We strongly recommend a study on a subset of the benchmark where both GPT-4.1 and human meteorology experts score the outputs, with a report on the correlation. This is essential to quantify the judge's reliability.

Q2: Please discuss the potential impact of known LLM judge biases (e.g., verbosity, reasoning style) on your results. Could you design an experiment to test for these biases?

Q3: We urge you to expand the "Related Work" section to include and differentiate your work from recent geospatial dataset(like WeatherQA [1], ClimateIQA [2] and Terra [3]) and multimodal forecasting systems (like CLLMate[4]). We suggest adding a comparison table.

[1] Ma, C., Hua, Z., Anderson-Frey, A., Iyer, V., Liu, X., & Qin, L. (2024). Weatherqa: Can multimodal language models reason about severe weather?. arXiv preprint arXiv:2406.11217.
[2] Chen, J., Zhou, P., Hua, Y., Chong, D., Cao, M., Li, Y., ... & Yuan, Z. (2025, August). ClimateIQA: A New Dataset and Benchmark to Advance Vision-Language Models in Meteorology Anomalies Analysis. In Proceedings of the 31st ACM SIGKDD Conference on Knowledge Discovery and Data Mining V. 2 (pp. 5322-5333).
[3] Chen, W., Hao, X., Wu, Y., & Liang, Y. (2024). Terra: A multimodal spatio-temporal dataset spanning the earth. Advances in Neural Information Processing Systems, 37, 66329-66356.
[4] Li, H., Wang, Z., Wang, J., Wang, Y., Lau, A. K. H., & Qu, H. (2024). CLLMate: A Multimodal Benchmark for Weather and Climate Events Forecasting. arXiv preprint arXiv:2409.19058.

Q4: Can you provide more qualitative case studies of EWE's analysis of different event types (e.g., tropical cyclones, heatwaves) to better showcase its capabilities and limitations?

---

### Official Review · Reviewer_r1xr · 2025-10-31

**Soundness:** 2
**Presentation:** 2
**Contribution:** 2
**Rating:** 4
**Confidence:** 3

**Summary:**

This paper introduces the Extreme Weather Expert (EWE), an agentic framework designed to automate the diagnostic analysis of extreme weather events. The authors posit that this task, traditionally a labor-intensive process reliant on human experts, represents a significant bottleneck in climate science. The EWE framework aims to emulate expert workflows by integrating three primary components: (1) Knowledge-Enhanced Planning, which uses expert-annotated Chain-of-Thought (CoT) exemplars to structure the agent's reasoning; (2) Self-Evolving Closed-Loop Reasoning, an iterative cycle of thought, action, observation, and interpretation that includes self-debugging and a dual auditor module for code and content verification; and (3) a specialized Meteorological Toolkit with functions for data retrieval and domain-specific computations.

**Strengths:**

**Problem Significance:** Automating the diagnostic analysis process, as proposed, could dramatically accelerate research, improve forecasting models, and inform climate adaptation strategies, particularly in developing nations that often lack dedicated meteorological expertise. The motivation for this work is compelling and well-articulated.

**Novel Benchmark Dataset:** The curation of a benchmark dataset of 103 high-impact extreme weather events is a significant and tangible contribution. This dataset, sourced from reputable databases like EM-DAT and WMO reports and annotated with key metadata, provides a much-needed resource for standardised evaluation and future research in this nascent subfield.

**End-to-End System Vision:** The paper presents an ambitious attempt to construct a complete, end-to-end system that progresses from a high-level user query to a detailed, multi-faceted analytical report.

**Weaknesses:**

**Relation to the ReAct Framework:** The core reasoning loop of "Thought, Action, Observation, and Interpretation" is functionally very similar to the established ReAct paradigm. Explicitly framing the work as an application and domain-specific adaptation of ReAct would help clarify the paper's contribution, shifting the focus from the framework's structure to its successful implementation in a complex scientific domain.

**Contextualizing Tool Use:** The "Meteorological Toolkit" is a necessary component, but the paper could benefit from a brief discussion of more advanced tool-use paradigms, such as the self-supervised tool acquisition in Toolformer. This would help contextualize the design choice of using a curated, static toolkit and highlight potential avenues for future work.

**Acknowledging Prior Work in Scientific Agents:** The claim of being the "first" agent for scientific analysis could be nuanced. Agentic frameworks have been explored in other scientific domains, with ChemCrow in chemistry being a notable example. Positioning EWE in relation to such systems would create a more compelling narrative about the shared challenges and domain-specific solutions for building scientific agents, strengthening the paper's impact on the broader field.

**Addressing "LLM-as-a-Judge" Biases:** The evaluation relies on a single MLLM (GPT-4.1) as a judge. The literature has well-documented concerns about the reliability of this approach, citing susceptibility to positional, verbosity, and other cognitive biases that can affect reproducibility. The paper would be stronger if it acknowledged these potential limitations and incorporated mitigation strategies, such as using multiple judge models to measure inter-rater reliability or performing a small-scale calibration against human expert scores.

**Justifying the Choice of Evaluator:** The paper dismisses Gemini-2.5-Pro as an evaluator because it "tended to penalize correct code". This justification would be more convincing if supported by a small-scale comparative analysis against ground-truth human scores. Providing this data would strengthen the choice of GPT-4.1 as the sole judge and increase confidence in the reported results.

**Expanding the Ablation Study:** The authors note that the ablation study was conducted "on a limited set of samples". While the initial results are informative, expanding this analysis to a larger subset of the benchmark would provide more conclusive evidence for the contribution of each framework component.

**Completing the Ablation Design:** The current ablation study (Table 2) does not include a condition that isolates the contribution of Chain-of-Thought (i.e., ✓ Tools, ✓ Auditor, X CoT). Including this condition would allow for a more direct and powerful conclusion about the foundational role of CoT in the framework.

**Incorporating a Human Baseline:** The framework is designed to "emulate expert workflows". A small-scale comparison against analyses produced by human experts, even on a few benchmark events, would be highly valuable. This would provide a crucial baseline to ground the agent's performance and help calibrate the automated evaluation metrics.

**Questions:**

**Regarding Novelty:** The proposed agentic loop of "Thought, Action, Observation, Interpretation" appears to be a direct application of the ReAct framework (Yao et al., 2023). Could you please clarify what specific, novel contributions your framework makes to the theory or practice of agentic architectures beyond applying the ReAct paradigm to a new scientific domain?

**Regarding Evaluation Rigor:** Your entire quantitative evaluation rests on a single MLLM judge (GPT-4.1). Given the extensive literature on LLM judge biases (e.g., positional, verbosity, and leniency biases), how can you assure the community that the results in Table 1 are reliable and reproducible? Could you please provide the exact prompts and rubrics used for the judge, and a more scientific justification for dismissing Gemini-2.5-Pro as an evaluator beyond the anecdotal claim that it "penalize[d] correct code"?

**Regarding "Self-Evolving":** The term "Self-Evolving" implies a learning or adaptation process that persists beyond a single run. However, the mechanism described appears to be iterative self-correction within a single task. Could you please clarify if the agent's strategies or parameters are updated and persist across different tasks? If not, would you consider a more precise term like "Iterative Refinement"?

**Regarding Auditor Implementation:** The "Content Auditor" is described as assessing subjective qualities like "perceptual clarity." Could you provide the specific implementation details, including the prompts and any quantitative metrics used, to demonstrate how this process is made objective and repeatable?

**Regarding Ablation Studies:** The ablation study was performed on a "limited set of samples" and is missing a key condition to isolate the effect of Chain-of-Thought (✓ Tools, ✓ Auditor, X CoT). Can you justify why these experimental limitations do not undermine the conclusions drawn about the importance of each component of the EWE framework?

**Regarding Scientific Reasoning:** Real-world scientific analysis often involves dealing with uncertainty, ambiguity, and conflicting data sources. How does the EWE framework handle scenarios where different diagnostic tools or data visualisations suggest contradictory physical mechanisms? Is there a mechanism for expressing uncertainty or confidence levels in its final report?

#### References

- ReAct: Synergizing Reasoning and Acting in Language Models. [ICLR 23]
- Toolformer: Language Models Can Teach Themselves to Use Tools. [NeurIPS 23]
- ChemCrow: Augmenting large-language models with chemistry tools. [NeurIPS2023-AI4Science]
- Agentic AI for Scientific Discovery: A Survey of Progress, Challenges, and Future Directions. [arXiv:2503.08979]
- Judging LLM-as-a-Judge with MT-Bench and Chatbot Arena. [NeurIPS 2023]
- Evaluating Scoring Bias in LLM-as-a-Judge. [arXiv:2506.22316]

---

### Official Review · Reviewer_38nD · 2025-10-31

**Soundness:** 1
**Presentation:** 1
**Contribution:** 1
**Rating:** 2
**Confidence:** 5

**Summary:**

The authors propose an agent framework for automated diagnostic reasoning of extreme weather events. While the topic might be timely, the paper is in a poor representation, lacking of clarity and scientific validation. A lot of details of the framework are missing.

**Strengths:**

Automated diagnostic reasoning of extreme weather events could be an important research topic.

**Weaknesses:**

1. The details of the method are missing. For example, the authors mentioned embedded meteorological knowledge, but they never explained what kind of knowledge or rules are embedded, how they annotate for the CoT guidelines,
2. The dataset they created is one of the main contributions as the authors wrote. However, they did not provide the details of the dataset, such as, how they picked up 103 extreme events, how many variables did they select, etc.
3. The evaluation in this paper is not reliable, even incorrect. The authors use a MLLM to evaluate a LLM-based workflow, how do the authors make sure the MLLM evaluator is correct and robust? Evaluation of interpretation is very important, especially for those scientific domains. Using GPT-4.1 and Gemini-2.5-pro as judge is not reliable to me. Importantly, what the authors mentioned in (5.1 experimental settings, evaluation), “While we tested Gemini-2.5-Pro as an alternative, it tended to penalize correct code, leading to inaccurately low scores. Consequently, we designated GPT-4.1 as the sole judge for our evaluation to ensure the consistency and reliability of our results.”, actually indicating that using LLM as an evaluator is definitely not reliable, and should be avoided. This strongly undermines the credibility of their proposed workflow.
4. There is no evidence in the paper that the proposed workflow can help scientific discovery. The authors did not provide any information or case study how their framework can produce new scientific insights, confirm or correct experts’ analysis, or improve the model explainability.

**Questions:**

1. How do the authors make sure the LLM-based metric is faithful and access a LLM-based workflow appropriately?
2. Can the author explain how they manually annotate step-by-step analytical guidelines? Otherwise, it's difficult to understand what kind of guidelines that are provided to the agent later for the planning.
3. What analysis does the code auditor conduct, and how to make sure the auditor correctly access the code?
4. What is content auditor, how does it ensure the semantic integrity and perceptual clarity?
5. Can the authors specify what functions, how many functions in the curated library they mentioned?
6. Can the authors clarify the exemplar, and how's the generated checklists like?
7. The authors mentioned they have a specific behavioral category. How many behavior classes do you have? Can the authors clarify the step classification task clearly?
8. Regarding the SG and CG evaluation for the 7 stages. Can the authors discuss the evaluation in detail? In current version, I can't understand what's the criteria for these 7 stages, how to grade them, what's the initial scores before normalization to [0,1].

---

### Meta-Review · Area_Chair_xPG8 · 2026-01-07

**Summary:**

The paper attempts to address an important problem; however, it lacks sufficient in-depth discussion of its technical details and concrete innovations. The empirical evaluation is also not sufficiently convincing.

**Reviewer Concerns:**

The author did not post a rebuttal, so the reviewers' concerns are not addressed.

**Reviewer Scores:**

There is no feedback posted from the author. I do not expect the reviewer to receive any additional information that would change their score.

---

### Decision · Program_Chairs · 2026-01-26

Reject